Identity-based linear homomorphic signature for a restricted combiners’ group for e-commerce

Tian Yuan 1
Song Weitao 1 weitaosong@163.com
Zhou Tanping 2 tanping2020@iscas.ac.cn
Hu Bin 1
http://orcid.org/0009-0004-0509-6969 Zhou Xuan 2
Ding Yujie 2
Zhong Weidong 2
Yang Xiaoyuan 2
1 Information Engineering University , Zhengzhou, Henan , China
2 Chinese People’s Armed Police Force Engineering University , Xi’an, Shaanxi , China
Akleylek Sedat
Electronic publication date: 2025 Aug 4
Publication date: 2025
Volume: 11
Electronic Location ID: e3068
Received 2023 Jun 3; Accepted 2025 Jul 3
Copyright: © 2025 Tian et al.
Copyright year: 2025
Copyright holder: Tian et al.
License: This is an open access article distributed under the terms of the Creative Commons Attribution License, which permits unrestricted use, distribution, reproduction and adaptation in any medium and for any purpose provided that it is properly attributed. For attribution, the original author(s), title, publication source (PeerJ Computer Science) and either DOI or URL of the article must be cited.
License URL: https://creativecommons.org/licenses/by/4.0/

Keywords: Homomorphic signature, Identity-based linear homomorphic signature, Network coding

Funding: National Cryptologic Science Fund of China 2025NCSF02044 National Key R&D Program of China 2023YFB3106100 National Natural Science Foundation of China 62172436, 62102452 Natural Science Foundation of Shaanxi Province 2023-JC-YB-584 This work was supported by the National Cryptologic Science Fund of China (2025NCSF02044), National Key R&D Program of China (2023YFB3106100), National Natural Science Foundation of China (62172436, 62102452) and Natural Science Foundation of Shaanxi Province (2023-JC-YB-584). The funders had no role in study design, data collection and analysis, decision to publish, or preparation of the manuscript.

==============================
As the volume of electronic transaction data increases and the demand for real-time processing grows, network coding techniques have become popular for improving performance. However, early implementations often overlooked critical data security issues, such as forgery and data leakage. While existing homomorphic signature schemes effectively ensure data integrity, they can unintentionally allow malicious actors to exploit intermediate signatures. This misuse can lead to unnecessary bandwidth consumption and hinder the verification processes for legitimate users. To address the problem of malicious combinations, we apply the Chinese Remainder theorem (CRT) to establish a layer of secret-sharing that restricts access to authorized users in conjunction with homomorphic signatures. Furthermore, we introduce a formal definition for an identity-based linear homomorphic signature for a restricted combiners’ group (IBLHS-RCG). This framework integrates linear homomorphic signatures with the CRT within the context of e-commerce, enabling us to develop a specialized scheme for IBLHS-RCG. We demonstrate that our scheme is unforgeable against adaptive chosen-message attacks. Additionally, simulations conducted using the Python Pairing-based Cryptography Library (PYPBC) show that the signing and verification costs of our approach are low.

Introduction

The security of electronic commerce (Al-Zubaidie, Zhang & Zhang, 2020; Liu et al., 2023) is vital for both individuals and the broader economy. According to the 52nd Statistical Report on China’s Internet Development and data from the National Bureau of Statistics of China, the number of electronic payment users in China reached 943 million by June 2023, marking an increase of 911 million since December 2022. This represents 87.5% of the total number of internet users and 66.9% of the national population. As illustrated in Fig. 1, electronic commerce (Cao, 2023; Al-Zubaidie & Shyaa, 2023) consists of three primary components: service platforms, server clusters, and user groups. The service platform is connected to various servers through a wired network, while each server cluster wirelessly broadcasts to surrounding areas, enabling user access via mobile devices. Different server clusters, such as enterprise and personal transaction databases, store specific data assigned by the service platform and provide validation services for distinct user groups. As the Internet becomes the dominant arena for electronic commerce, ensuring transaction security has emerged as a critical concern. Hackers can exploit network vulnerabilities to compromise systems, resulting in the theft of funds or personal financial information. Moreover, insecure electronic commerce practices can lead to legal challenges and a significant erosion of user trust.

Figure 1 Architecture of electronic services.

Signature schemes that integrate secure multi-party computing (SMPC) are an effective security mechanism for ensuring transaction legitimacy and security. However, mitigating the risk of malicious combinations of intermediate signatures by combiners remains a substantial challenge. To address this issue, we propose a twofold approach: first, designing a signature scheme that incorporates secret segmentation techniques; second, employing homomorphic signature algorithms in place of traditional digital signature algorithms. Homomorphic signatures offer advantageous properties such as secure transmission and verifiable, yet unsolicited, information, effectively mitigating potential information leakage during data transmission. Since e-commerce clients primarily utilize mobile devices, such as cell phones, which have limited capacity and lower computational efficiency, we will implement a linear homomorphic signature algorithm characterized by short signature lengths and high signing and verification efficiency.

In practical electronic commerce (Liu et al., 2023), our scheme is applicable to multiparty signature scenarios, including multiparty account management, transaction authorization, and payment processing. For instance, in a tri-party transaction, one party can generate a signature using their private key while utilizing a shared session key to authorize the inclusion of secondary signers. Additionally, the identity and permissions of each signer can be restricted and managed based on specific needs and roles, thereby ensuring the security and reliability of the transaction.

Our contributions

Our proposed scheme, identity-based linear homomorphic signature for a restricted combiners’ group (IBLHS-RCG), restricts the combiners’ group, proves it is correct and secure in the random oracle model, and operates efficiently under the computer simulation experiments. The specific contributions of this article are as follows: 1. We propose a novel linear homomorphic signature (LHS) scheme that restricts specific authorized intermediate nodes from performing linear combinations of signature vectors according to different tasks. This measure effectively prevents unauthorized nodes from maliciously combining signature vectors.

2. We construct the first linear homomorphic signature scheme that is simultaneously resistant to malicious combinations, based on the original designated single combiner signature scheme (Lin, Xue & Huang, 2021), we expanding its functionality to a linear homomorphic signature scheme that can specify multiple combiners.

3. We compress the time for signature and signature verification to sublinear complexity. Our scheme’s achieved signature length does not change after combination, and it reduces the verification and communication costs during transmission and after the combination signature more than Lin’s designated combiner scheme (Lin, Xue & Huang, 2021).

4. We propose an identity-based linear homomorphic signature scheme for restricted combiner group. we combine this article’s scheme with the identity-based signature scheme, which makes the scheme more compatible with the IP network environment.

Related work

Electronic transactions (Cao, 2023; Al-Zubaidie & Shyaa, 2023) must occur in environments characterized by low latency and well-defined authorization protocols. Various external defense mechanisms and content-based security strategies have been proposed to enhance network protocols. Among these, network coding facilitates efficient content distribution while maximizing transmission throughput. However, a notable drawback is that data packets within network coding are susceptible to corruption, which presents significant challenges for secure data transmission.

To protect data confidentiality, homomorphic encryption can be employed. While encryption prevents attackers from eavesdropping on the original data, it does not inherently verify data integrity and authenticity. This gap can have serious ramifications in financial transactions, as tampered transaction data during transmission can impair recognition by the combiner group, thereby undermining the credibility of electronic transactions and causing substantial harm to electronic finance.

Thus, in addition to securing data content, ensuring data integrity and authenticity is crucial. One effective approach is the use of digital signatures combined with message digest functions to verify both the authenticity and integrity of data packets. Digital signatures provide non-repudiation, ensuring that the sender cannot deny the transmission of a specific message, while message digest functions guarantee that the message remains intact throughout its transmission. The homomorphic signature scheme represents a promising solution that simultaneously addresses these essential criteria.

In the era of big data, with increasing awareness of privacy protection, more attention is being given to securing data during network transmission and ensuring stored data integrity in the cloud. While digital signatures guarantee data integrity and verify data owner identity, their lack of homomorphic properties leads to increased bandwidth consumption during transmission—a major obstacle to their deployment in network environments. Homomorphic signatures aim to preserve the benefits of digital signatures while enabling operations on smaller, homomorphically linked packets. This leverages the aggregation capabilities of intermediate nodes, reducing both the number of packets transmitted and their size, thereby improving efficiency.

Early in 2000, Rivest proposed the basic concept of homomorphic signature schemes (Rivest, 2000). In 2002, the first formal definition of homomorphic signatures was proposed by Johnson et al. (2002). After 2002, research achievements in homomorphic signature schemes emerged in succession, such as linear homomorphic signature (LHS) schemes (Lin, Xue & Huang, 2021; Boneh et al., 2009; Boneh & Freeman, 2011b; Gennaro et al., 2010; Lin et al., 2018; Wang, Hu & Wang, 2013), polynomial homomorphic signature (PHS) schemes (Boneh & Freeman, 2011a), fully homomorphic signature (FHS) schemes (Gentry, 2009; Wang et al., 2015; Wang & Wang, 2020), homomorphic aggregate signature (HAS) schemes (Jing, 2014; Zhang, Yu & Wang, 2012), and multi-key homomorphic signature (MHS) schemes (Aranha & Pagnin, 2019; Fiore et al., 2016; Lai et al., 2018). Among them, LHS schemes (Zhao et al., 2007; Yu et al., 2008; Boneh et al., 2009; Cheng et al., 2016; Li et al., 2018; Chang et al., 2019; SadrHaghighi & Khorsandi, 2016; Zhang et al., 2018; Li, Zhang & Liu, 2022) are particularly useful in preventing pollution attacks in linear network coding.

In 2007, Zhao et al. (2007) proposed a linear homomorphic signature scheme with a network encoding function, which implements a signature scheme in a network encoding environment at a negligible cost; however, its security cannot be guaranteed. In 2008, Yu et al. (2008) proposed an antipollution homomorphic signature scheme based on the RSA difficulty problem, which can achieve improvements in efficiency and security. After 2009, Boneh et al. (2009) proposed the first formal definition of linear homomorphic signatures and proposed more efficient and secure signature schemes in network environments. At this point, research on linear homomorphic signatures began to move in the right direction. Based on Boneh et al. (2009), many linear homomorphic signature schemes based on public key infrastructure (PKI) were designed (Cheng et al., 2016; Li et al., 2018). However, these schemes are unsuitable for electronic transactions with high time requirements because the signature of the data packet needs to be bound with corresponding certificates, resulting in the signature length of the data packet being much longer than the data packet. Subsequently, some identity-based linear homomorphic signature schemes (Chang et al., 2019; Zhang et al., 2018) were proposed to address this issue. These schemes reduce the length of the signature, but the excessive signature generation and verification costs still cannot satisfy the requirements of the Internet environment. It was not until 2022 when Li, Zhang & Liu (2022) proposed a secure and efficient identity-based homomorphic signature, that the cost of signature generation and verification began to level out. For a comprehensive comparison of the various types of linear homomorphic signature schemes, the nature of the underlying hard problems, and other relevant factors, please refer to Table 1.

Table 1 Comparison of homomorphic signature schemes.

Year	Authors	Scheme type	Hard problem	
2007	Zhao et al.	LHS	(p, k, n)-Diffie-Hellman problem	
2008	Yu et al.	LHS	Integer factorization problem	
2009	Boneh et al.	the formal definition of LHS	Co-computational Diffie-Hellman problem	
2016	Cheng et al.	PKI-based LHS	Integer factorization problem	
2018	Li et al.	PKI-based LHS	Short integers solutions problem	
2018	Zhang et al.	ID-based LHS	Co-computational Diffie-Hellman problem	
2019	Chang et al.	ID-based LHS	Computational Diffie-Hellman problem	
2022	Li et al.	ID-based LHS	Co-computational Diffie-Hellman problem	
Note:

Zhao et al. (2007), Yu et al. (2008), Boneh et al. (2009), Li et al. (2018), Zhang et al. (2018), Chang et al. (2019), Li, Zhang & Liu (2022).

However, beyond ensuring transaction verifiability, electronic transactions should also exhibit partial transparency rather than full transparency. This is inherent to the privileged service model of electronic transactions, where only specific, partially privileged intermediate nodes can perform combination and verification at each step. Direct application of homomorphic signatures, however, would render the entire transaction process fully transparent across the network—an outcome we seek to avoid. After combining Li, Zhang & Liu (2022) and Lin, Xue & Huang (2021), we propose an efficient signature scheme that can specify multiple combiners and maintain the generation and verification costs of signatures at a constant level.

Article structure

The rest of this article is structured as follows. In the “Preliminaries and Related Definitions” section, we briefly review some basic concepts in linear homomorphic signature schemes. In the “Concrete Scheme” section, we present the proposed IBLHS-RCG scheme and analyze its correctness. In the “Security Analysis” section, we provide security proof of the scheme. In the “Performance Analysis” section, we provide a performance evaluation of the scheme. Finally, we provide our conclusions in the “Conclusions” section.

Preliminaries and related definitions

This section briefly introduces the proof model used in this scheme, the theoretical basis required for construction, and the formal definition of this scheme.

Preliminaries

Existential unforgeability under adaptive chosen message attacks (EUF-CMA) security model (Boneh & Freeman, 2011a). If an adversary of a probabilistic polynomial time algorithm (PPT) cannot win the challenge response game with advantage ϵ within time t after asking for q signatures, then a signature scheme is said to have achieved (t,q, ϵ)-security in the EUF-CMA security model. That is, the adversary satisfies the following properties: 1. The adversary can forge the signature σ∗ for v∗.

2. The forged σ∗ has not been queried during the interrogation phase.

Asymmetric bilinear mapping (Boneh & Franklin, 2001; Verheul, 2001; Boneh, Lynn & Shacham, 2004; Miller, 2004). Select three cyclic groups G1,G2,GT, where ord(G1)=ord(G2)=ord(GT)=p ( p is a prime number), and define a mapping e:G1×G2→GT that satisfies the following relationship: 1. Bilinear: ∀g∈G1,h∈G2,a,b∈Zp, we have e(ga,hb)=e(g,h)ab.

2. Nondegeneracy: ∀g∈G1\{1},h∈G2\{1},e(g,h)≠1GT,1GT is the identity element of GT.

3. Computability: ∀g∈G1,h∈G2,e(g,h) can be efficiently calculated.

Co-Computational Diffie-Hellman (co-CDH) problem (Boneh, Lynn & Shacham, 2004; Huang & Wang, 2010; Galbraith & Verheul, 2008). We define two bilinear groups G1 and G2, where ord(G1)=ord(G2)=p ( p is a prime number); we randomly select a∈Zp, give g∈G1,h,ha∈G2, and the algorithm A can solve ga. We denote this probability as Pr(PPTA(g,h,ha)=ga,a∈Zp).

Co-CDH assumption (Huang & Wang, 2010; Ng, Tan & Chin, 2018; Chang et al., 2017). For any PPT adversary, the probability that they can solve the co-CDH problem is negligible; that is, Pr(PPTA(g,h,ha)=ga,a∈Zp)=negligible, meaning that the PPT adversary wanting to solve the co-CDH problem will have difficulty.

Chinese Remainder theorem (CRT) (Schindler, 2000; Zheng, Huang & Matthews, 2007; Quisquater, Preneel & Vandewalle, 2002). Suppose that m1,m2,…,mn are mutually prime for any two pairs. The equations

(1) {x≡a1(modm1),x≡a2(modm2),⋮x≡an(modmn)

have a solution. If χ1,χ2,… are the solutions to these equations, then we have χ1≡χ2≡⋯≡x(modM), M=∏i=1nmi,x≡∑i=1nai⋅Mmi⋅(Mmimodmi)−1modM.

ID-based signature

An identity-based signature scheme typically consists of four algorithms: initialization algorithm Setup, key extraction algorithm KeyExt, signing algorithm Sign and verification algorithm Verify. The process involves interaction between the private key generator (PKG) and the user. The main steps are as follows: (params,msk)←Setup(1λ): Input security parameter 1λ, output public parameters params and master secret key msk, where msk is kept secret by the PKG.

skID←KeyExt(msk,ID): Input master secret key msk and user’s identity ID, PKG computes the user’s private key skID, and transmits it to the user via a secure channel.

σ←Sign(skID,m): Input the user’s private key skID and the message m to be signed, output the user’s signature σ for the message m.

0/1←Verify(ID,params,m,σ): Input the user’s identity ID, the public parameters params, the message m and its signature σ, if the verification is passed, output 1; otherwise, output 0.

The formal definition of LHS

A linearly homomorphic signature scheme (LHS) (Boneh et al., 2009) consists of the following four PPT algorithms.

(PK,SK)←Setup(1λ,N): Input the security parameter λ and an integer N, generate the bilinear group G=(G1,G2,GT,p,e,φ), choose the generator element h of G2 and randomly choose an element α on Fp∗, such that u=hα, H:{0,1}∗×{0,1}∗→G1, output the public key PK=(G,H,h,u) and the private key SK=α.

σ←Sign(SK,id,v): Input private key SK, file label id={0,1}λ and vector v=(v1,⋯,vn), output signature σ=(∏i=1NH(id,i)vi)α.

σ←Combine(PK,id,{βi,σi}i=1l): Input public key PK, file label id, and binary tuple {βi,σi}i=1l, and then output signature σ=∏i=1lσiβi.

0/1←Verify(PK,id,y,σ): Input public key PK, file label id, verification vector y=(y1,⋯,yn), and signature σ. If e(σ,h)=e(∏i=1NH(id,i)yi,u) is established, the verification passes and outputs 1; otherwise, outputs 0.

Security definitions

In this subsection, we will detail the properties and definitions (Boneh & Freeman, 2011a; Ateniese et al., 2000) related to security analysis as follows.

Unforgeability

Definition 1. The security can be defined using the following game between adversary Adv and challenger chal. Note that both chal and Adv cannot know both of user’s private key and the user’s access structure. 1. Initialization phase: The chal generates public parameters pp and system master private key msk, then sends pp to Adv, and keeps msk to itself.

2. Random oracle querying phase: Adv obtains favorable information by randomly querying the chal, thus forging a non-existent message signature pair (M∗,σ∗) and passing the verification.

3. Outputs: When chal receives (M∗,σ∗), chal needs to use this forged message signature pair to solve the hard problem.

We need to ensure that the probability of chal solving the hard problem in Outputs is negligible.

Definition 2. A signature scheme is said to be (t,q,ϵ) security under the EUF-CMA security model if an adversary of a probabilistic polynomial time (PPT) algorithm is still unable to win the challenge response game with an advantage ϵ in time t after querying the signature q times. That is, the challenge response game can be defined as follows: 1. Setup: Let params be a system parameter, where the challenger chal generates the key pair (pk,sk) from params. The challenger chal then sends pk to the adversary Adv, and chal retains sk for answering Adv’s signature queries.

2. Query: Adv adaptively selects the message mi for signature lookup, and chal generates the corresponding σi to return to Adv.

3. Output: Adv generates the message signature pair (m∗,σ∗), if: σ∗ is a legal signature for m∗.

All signatures on m∗ not queried.

we assume that Adv won the challenge response game.

Definition 3. If the (id∗,m∗,σ∗) generated by adversary in Output can make Verify (PK,id∗,m∗,σ∗) output 1, and if for ∀idi, id∗≠idi. We consider adversary to have won the game.

Confidentiality

Definition 4. This security can be defined as a game between Adv and chal. In this game, Adv can obtain only the message-signature pair (m∗,σ∗) from chal. If Adv derives information beyond the identity ID by inverse solving (m∗,σ∗), Adv wins the game, and the scheme fails to achieve confidentiality.

Unlinkability

Definition 5. This security can be defined as a game between Adv and chal. In this game, Adv randomly chooses 0 or 1 from {0,1} (Here it is assumed that 1 means (m1∗,σ1∗) and (m2∗,σ2∗) come from the same group of combiners, and 0 means (m1∗,σ1∗) and (m2∗,σ2∗) come from a different group of combiners). After n queries, if the probability of Adv guessing correctly is greater than 12, then the scheme does not have unlinkability.

Traceability

Definition 6. For the receiver Receiver, the receiver is able to verify the legitimacy of the signature σ by using the signer’s public key pkID, so that we consider the scheme to have Traceability.

Coalition-resistance

Definition 7. Assuming that any combiner without the server’s permission obtains Signer’s message signature pairs (m1,σ1) and (m2,σ2), we consider the scheme is not resistant to a coalition attack if the two combiners can directly generate a new efficient message-signature pair (m∗,σ∗).

Concrete scheme

In this section, we provide the details of our scheme. Our scheme introduces an innovative structure—the restricted combiner’s group. The restricted combiner’s group is primarily based on CRT, which converts the single key of Lin’s scheme into a sub-session key derived from a master session key via secret splitting. This enables the structural extension from the single combiner to the restricted combiner’s group. The design improves system flexibility and scalability, providing a robust foundation for subsequent security protocol development. First, our scheme, initialized by private key generation (PKG), generates corresponding public parameters. Second, all users (including signers, combiners, and receivers) need to submit their identity information to PKG to generate their own public keys and private keys. Then, the group of combiners requests the combination rights from the signer. After the signer approves the request, the signatures are distributed. After layer-by-layer verification, the group of combiners composes a signature combination network. Finally, the network provides services to the receiver. Figure 2 shows the process of data transmission and reception between two entities, and Fig. 3 shows the entire workflow of our scheme. In subsection “Correctness”, we demonstrate the correctness of our scheme. The specific details and relevant proof of this scheme are described as follows.

Figure 2 Process of data transmission and reception between two entities.

Figure 3 Entire workflow of our scheme.

The formal definition of IBLHS-RCG

Before presenting the formal definition of this scheme, we give the concrete notation Table 2 for a clearer understanding of our scheme. IBLHS-RCG is a linear homomorphic signature scheme that can only be combined in a specified group of combiners. Based on the original LHS signature, we add four steps KeyExt, Quest, GroupVerify, and PartialVerify, so that the combiners in the scheme cannot take up bandwidth by arbitrarily combining signatures. Our scheme consists of the following eight PPT algorithms. (params,msk)←Setup(1λ,N): This step is run by PKG, with the input of security parameter λ and augmented vector dimension N. The algorithm generates public parameter params and broadcasts it to the entire network while secretly preserving the master private key msk.

Table 2 The notations and meanings.

Notation	Meaning	
λ	Security parameter	
N	Augmented vector dimension	
msk	Master secret key	
uskID	User’s secret key (associated with ID)	
Q	Hierarchical parameter	
bID	Session sub-key (associated with ID)	
qID	Access level parameter (associated with ID)	
vk	Vector k	
B	Session sub-key	
σk	Signature corresponding to vector k	

uskID←KeyExt(ID,msk): This step is run by PKG, where the user inputs their own identity ID, PKG inputs msk, and the algorithm outputs the corresponding user’s private key uskID, which is returned to the user. After receiving uskID, the user can verify the credibility of the PKG.

(Q,bID)←Quest(ID,qID): This step is run by the signer and the combiners. The applicant combiners submit the identity ID and access level parameter qID to the signer to apply for the signature combination right of a certain quantum space V. The signer reviews whether the access level of the combiners has reached the security level V. If the access right application has passed, the signer returns (Q,bID) to the designated combiners.

({σk}k∈l,τ)←Sign(ID,uskID,{vk}k=1m,B,Q): This step is run by the signer, who splits and augments the packet data, signs the basic vectors, and adds the hierarchical parameter Q and session key B. The basic vector-signature pairs are divided into multiple packets and distributed to each designated combiner.

{0,1}←GroupVerify({qd,bd}d∈t,Q): This step is run by a designated group of combiners, and this group uses the CRT to restore the session master key B. If recovery is correct, the algorithm outputs 1; else, the designated combiners terminate the algorithm and output 0.

{0,1}←PartialVerify({ςk}l,d): This step is run by a designated group of combiners. In this step, the designated group of combiners can use the session master key B obtained from the algorithm GroupVerify to verify the correctness of the signature, thereby verifying the credibility of the signer.

(v~d,S~d,B′)←Combine({vk,ςk,ck}l,d): This step is completed independently and run by each combiner. Each combiner can combine their received basic vector-signature pairs to generate secondary vector-signature pairs. Then, each combiner sends the pairs to the receiver.

{0,1}←Verify(v~d,S~d): After receiving the secondary vector-signature pairs (v~1,S~1),(v~2,S~2),…, sent by the combiners, the receiver can verify the credibility of each combiner through pairing operations.

In our scheme, if any of the first four steps cannot be honestly executed or there are unauthorized combiners in the combiner group, algorithm GroupVerify will output 0, and the dishonest behavior of the signer will result in a return of 0 for the algorithm PartialVerify, while a dishonest operation in any step will make the outputs of algorithm Verify 0. Therefore, our scheme ensures the security of the signature and the designated combination by preventing any malicious behavior of the signer or combiners from passing the verification.

Concrete structure

In our scheme, the combiners need to use the signer’s ID to verify the signature once it has been received. If the received ID is incorrect, the signature cannot pass the verification. Otherwise, each signature scheme will default to make the signer’s ID a part of the signature. Therefore, we can distinguish a specific signer by extracting their ID information from their signature. The overall process of our scheme is shown in Figs. 3 and 4, and the steps are as follows.

Figure 4 Simplified workflow of our scheme.

1. System initialization: Setup

In this step, PKG generates the public parameter params and master private key msk based on the security parameter λ and positive integer N (where N is the dimension of the augmented vector). The details are as follows: (a) PKG generates three cyclic groups G1,G2,GT with the same prime order p(p>2λ), which satisfies an asymmetric bilinear mapping e:G1×G2→GT and φ:G2→G1.

(b) PKG randomly selects generators g∈G1,h∈G2 and then randomly selects a value s∈Zp∗ as msk, after which PKG calculates hs and uses the value as the PKG master public key mpk.

(c) PKG selects four collision-resistant hash functions H0:{0,1}∗×G2→Zp∗,H1:{0,1}∗×ZN→G1,H2:{0,1}∗× ZN×{0,1}λ×G2→Zp,H3:ZpN×{0,1}λ→G1.

(d) PKG broadcasts params:=(p,G1,G2,GT,e,g,h,mpk,H0,H1,H2,H3) to the whole network and then secretly remains msk.

2. User registration: KeyExt

This step is composed of two parts. First, the Signer, Combiners, and Receiver request uskID from PKG. Second, users need to check the correctness of their uskID. (a) The Signer, Combiners, and Receiver send identity ID to PKG, which randomly selects a value r∈Zp∗. Calculate R=hr, x=r+s⋅H0(ID,R), and send (x,R) as uskID to Signer, Combiners and Receiver through a secure channel.

(b) The Signer, Combiners, and Receiver check if the equation e(gx,h)=e(g,R⋅mpkH0(ID,R)) holds. If it holds, save uskID locally, otherwise an error log will be created.

3. Combiners apply for permission: Quest

In this step, the signer generates the hierarchical parameter Q corresponding to the data and the session key bID of the approved combiner. The specific process is as follows: (a) The signer randomly generates the session master private key B∈Zp∗ and verifies whether (ID,qID) meets the confidentiality level of this data. If so, the signer generates the session key bID=B(modqID) of combiners and records the qID.

(b) When the maximum number of applicants or the deadline for application permission has been reached, the signer calculates Q:=∏ID∈listqID and sends (bID,Q) to the corresponding approved Combiners. For the convenience of description, it is assumed that the number of approved combinations is t and that the dth combination already has (bd,Q,qd),qID∈Zp∗).

4. Signature generation: Sign

This step is divided into two parts. First, the signer splits and augments the data packet. Then, the signer signs the basic vectors and distributes the basic vector-signature pairs to the approved combiners. The specific process is as follows: (a) The signer splits the data into m n-dimensional vectors {mk}k=1m(mk∈Zpn) and then augments each mk to vk:=(mk1,…,mkn,vk1,…,vkm)∈ZpN, where vki=1 holds only in the case of k=i, and the value of vki is always 0 in other cases. Here, we define the tensor space V:=span(v1,…,vm).

(b) The signer assigns the file identifier fid∈{0,1}λ for V, binds the signer information to V using τ=(fid,R), and then signs each basic vector vk with the signature σk, as follows:

(2) σ^k=∏i=1mH1(ID,i)vki⋅g∑j=1nHe2(ID,j,τ)mkj

(3) σk=σ^kx⋅He3(vk,ID)BmodQ.

Then, the signer sends ({σk}l,τ) to the designated combiners, where l represents the basic vector signature index received by the combiners. (For example, if Combiner1 receives (σ1,σ2), then the index is l={1,2}).

5. Validate the combined population: GroupVerify

This step is divided into two parts. First, each approved combiner uses the CRT to jointly solve B. Then, we verify the combination group. The specific process is as follows: (a) First, each combiner calculates Qd=Qqd. Second, we calculate Qd−1(Qd−1⋅Qd≡1modqd). Then, we calculate Bd≡bdQd−1QdmodQ. Finally, all combiners calculate B′=∑d=1tBdmodQ together.

(b) Each combiner checks whether bd≡B′modqd is true (i.e., determines whether B′ = B is true). If it is true, there are no malicious combiners in the combiner group. Otherwise, the algorithm terminates.

6. Partial open verification: PartialVerify

Owing to the hierarchical protection mechanism of data, open verification can only be carried out within the approved group of combiners in this step, the signer can encrypt signature information using a session key, and no arbitrary combiners can be allowed to verify. The workflow is divided into two parts, as follows: (a) Each combiner calculates {ςk}l,d={σk⋅H3(vk,ID)−B′modQ}l,d.

(b) The signer sends ciphertext to each combiner through broadcasting format. (Only the legal combiners who own the session key can decrypt ciphertext to signature information, others are not able to obtain the user’s information from the ciphertext.) Any combiner can verify whether any e(ςk,h)=e(σ^k,R⋅mpkH0(ID,R)) is valid. If it is true, the signer is trustworthy; otherwise, the algorithm terminates.

7. Signature aggregation: Combine

In this step, the combiners first linearly combine the basic vectors and then add the session key. The specific process is as follows: (a) Each combiner uses the encoding coefficient {ck}l to perform the following operation on their respective {(vk,ςk)}l so that v~d=∑k∈lckvk,ς~d=∏k∈lςkck.

(b) The combiners calculate S~d=ς~d⋅H3(v~d,ID)B′modQ and send (v~d,S~d) to the receiver through intermediate nodes.

8. Transaction validation: Verify

After receiving the combiners’ (v~d,S~d), the receiver first uses B′ to decrypt the signature and then verifies the correctness of the signature. The specific process is as follows: (a) The receiver uses B′ to solve for ς~d and uses the collected {(v~d,ς~d)}t to perform a combination calculation to obtain (data,ς), where ς is the signature for data.

(b) The receiver tests whether e(ς,h)=e(σ^,R⋅mpkH0(ID,R)) is valid. If it is, the verification has passed. Otherwise, the verification has not passed.

Correctness

In this section, we assume that all participants faithfully execute the above algorithms and that the correctness of the signature can be checked from the following aspects.

Lemma 1. For (s,mpk) pairs generated by Setup, the following conditions must be satisfied: 1. For ∀ςk, e(ςk,h)=e(σ^k,R⋅mpkH0(ID,R)) is established.

2. For ∀S~d, ∏d=1te(S~d,h)=e(σ^,R⋅mpkH0(ID,R)) is established.

3. For ∀ς, e(ς,h)=e(σ^,R⋅mpkH0(ID,R)) is established.

1. For the (vk,ςk) sent by the signer, we can conclude that e(ςk,h) satisfies the relationship mentioned in the previous section. The details are as follows:

(4) e(ςk,h)=e(σk⋅He3(vk,ID)−B′modQ,h)=e(σ^kx,h)=e(σ^k,hx)=e(σ^k,hr+s⋅H0(ID,R))=e(σ^k,R⋅mpkH0(ID,R)).

2. For the (v~d,S~d) sent by the above combiners, the receiver first calculates ς~d using the CRT, and the expression for the signature ς of data is as follows: (5) ς=∏d=1tς~d=∏d=1t∏k∈lςkck=∏d=1t∏k∈l(∏i=1m(H1(ID,i)vki⋅g∑j=1nH2(ID,j,τ)mkj)x)ck=∏d=1t∏k∈l∏i=1m(H1(ID,i)ckvki⋅g∑j=1nH2(ID,j,τ)ckmkj)x=(∏i=1mH1(ID,i)vi⋅g∑j=1nH2(ID,j,τ)mj)x.

(6) ∏d=1te(S~d,h)=e(∏d=1tS~d,h)=e(ς,h)=e((∏i=1mH1(ID,i)vi⋅g∑j=1nH2(ID,j,τ)mj)x,h)=e(∏i=1mH1(ID,i)vi⋅g∑j=1nH2(ID,j,τ)mj,hx)=e(∏i=1mH1(ID,i)vi⋅g∑j=1nH2(ID,j,τ)mj,hr+s⋅H0(ID,R))=e(∏i=1mH1(ID,i)vi⋅g∑j=1nH2(ID,j,τ)mj,R⋅mpkH0(ID,R))=e(σ^,R⋅mpkH0(ID,R)).

3. For the signature ς of the data, we can conclude that e(ς,h) satisfies the relationship mentioned in the previous section. The details are as follows:

(7) e(ς,h)=e((∏i=1mH1(ID,i)vi⋅g∑j=1nH2(ID,j,τ)mj)x,h)=e(∏i=1mH1(ID,i)vi⋅g∑j=1nH2(ID,j,τ)mj,hx)=e(∏i=1mH1(ID,i)vi⋅g∑j=1nH2(ID,j,τ)mj,hr+s⋅H0(ID,R))=e(∏i=1mH1(ID,i)vi⋅g∑j=1nH2(ID,j,τ)mj,R⋅mpkH0(ID,R))=e(σ^,R⋅mpkH0(ID,R)).

Security analysis

In this section, we will perform a comprehensive analysis of the security of our scheme, focusing on five key properties: unforgeability, confidentiality, unlinkability, traceability, and coalition-resistance. Detail are as follows.

Unforgeability proof

In this section, we prove that the scheme can achieve adaptive selection under the random oracle model for vector subspace, the existence of which is unforgeable under the attack of messages. Here, we use the challenge response mode to prove this. We define the challenger as chal, the PPT adversary as Adv, and the number of times Adv interrogates the random oracle i as Oi.

Initialization. chal let params′=(p,G1,G2,GT,e,g,h,mpk,H3), and we define mpk=Z, set the map φ:G2→G1.

Query the random oracle. The following seven query algorithms are adaptively selected between Adv and chal.

Create Query. The number of times that Adv asks chal’s IDi for the value of Ri is recorded as Oc, and the number of times that Adv creates a user query for ID as a challenger is recorded as I(I∈[1,Oc]). Whenever Adv asks IDi for Ri, chal performs the following steps: When i≠I, chal randomly chooses αi,βi∈Zp∗, and we define xi=αi,Ri=hxi⋅Z−βi. Then, chal returns Ri to Adv and stores (IDi,Ri,αi,βi) into list l0.

When i=I, chal randomly chooses αi,βi∈Zp∗, and we define xi=⊥,Ri=hαi. Then, chal returns Ri to Adv and stores (IDi,Ri,αi,βi) into list l0.

Ext Query. Whenever Adv queries chal’s IDi for the value of xi, chal performs the following steps: When i≠I, chal queries lc and returns αi to Adv.

When i=I, chal terminates the Ext Query algorithm.

H0 Query. Whenever Adv queries (IDi,Ri) for the value of H0, chal performs the following steps: If (IDi,Ri)∈lc, chal returns βi as the value of H0(IDi,Ri).

If (IDi,Ri)∉lc, chal runs the Create Query algorithm and then returns βi as the value of H0(IDi,Ri).

H1 Query. Whenever Adv queries vk for the value of {H1(ID0,i)}i=1m for the first time, chal performs the following steps: If ID0≠IDI, chal randomly chooses γ1,γ2,…,γm∈G1\{1}, and returns {γi}i=1m as {H1(ID0,i)}i=1m. Then, chal stores (ID0,{i}i=1m,{H1(ID,i)}i=1m) into list l1.

If ID0=IDI, chal randomly chooses δ1,ϵ1,δ2,ϵ2,…,δm,ϵm∈Zp∗, calculates {gδi⋅φ(h)ϵi}i=1m, and returns it as {H1(ID0,i)}i=1m. Then, chal stores (ID0,{i}i=1m,{H1(ID,i)}i=1m) into list l1.

When Adv queries other vectors for the H1 value of (ID0,{i}i=1m), chal queries whether (ID0,{i}i=1m) is on the list l1. If it is, chal returns {H1(ID0,i)}i=1m. Otherwise, chal runs the H1 Query algorithm.

H2 Query. Whenever Adv queries vk for the value of {H2(ID0,j,τ)}j=1n, chal randomly chooses ζ1, ζ2, …, ζn∈Zp, and returns {ζj}j=1n as {H2(ID0,j,τ)}j=1n. Then, it stores (ID0,{j}j=1n,τ,{H2(ID0,j,τ)}j=1n) into list l2.

When Adv queries other vectors for the H2 value of (ID0,{j}j=1n,τ), chal queries whether (ID0,{j}j=1n,τ) is on the list l2. If it is, chal returns {H2(ID0,j,τ)}j=1n. Otherwise, chal runs the H2 Query algorithm. Q Query. First, chal randomly chooses η∈Zpλ, and the number of times Adv asks chal’s IDi for the value of qi is denoted as Oq. When Adv runs the ith session key query, if the queried ID is equal to the chal’s ID, record this query as the Jth query (J∈[1,Oq]). Whenever Adv asks IDi for qi, chal performs the following steps: If i≠J, chal randomly chooses θi∈Zp∗, and calculates ϑi≡ηmodθi. Then, chal returns (θi,ϑi) as IDi’s (qi,bi) and stores (IDi,qi,bi) into list lq.

If i=J, chal terminates the Q Query algorithm.

Sign Query. When Adv asks for the signature of ID0’s basic vectors v1,v2,…,vm, chal performs the following steps: If ID0≠IDI, chal runs the following steps: - chal queries whether ID0 is in list l1. If it is, it extracts {γi}i=1m. Otherwise, chal runs the H1 Query algorithm before extracting.

- chal queries whether (ID0,τ) is in list l2. If it is, it extracts {ζj}j=1n. Otherwise, chal runs the H2 Query algorithm before extracting.

- chal calculates the signature σk of vk by the equation:

σk=(∏i=1mγivki⋅g∑j=1nζj⋅mkj)x0⋅H3(vk,ID0)ηmod∏i=1Oqθi.

- chal returns (τ,{σi}i=1m) to Adv.

If ID0=IDI, chal runs the following steps: - chal queries whether ID0 is in list l1. If it is, it extracts {gδi⋅φ(h)ϵi}i=1m. Otherwise, chal runs the H1 Query algorithm before extracting.

- chal queries whether (ID0,τ) is in list l2. If it is, it extracts {ζj}j=1n. Otherwise, chal runs the H2 Query algorithm before extracting.

- chal calculates the signature σk of vk, as shown in Eq. (8)

(8) (∏i=1m(gδiφ(h)ϵi)vki⋅g∑j=1nζj⋅mkj)xI⋅H3(vk,ID0)ηmod∏i=1Oqθi=φ(RI)∑i=1mϵivkiφ(Z)β0∑i=1mϵivki⋅H3(vk,ID0)ηmod∏i=1Oqθi

- chal returns (τ,{σi}i=1m) to Adv.

Outputs. We call Adv to win the game in the EUF-CMA security model if Adv forged signature σ∗ meets the following conditions: σ∗ is the legal signature of v∗.

ID∗=IDI.

The tensor space V∗ has not been asked for any signature by Adv.

Adv can pass both Partial open verification and Transaction validation.

First, Adv needs to be able to calculate η, and when Adv has sufficient numbers of (θi,ϑi), it can solve for the corresponding η. Below, we prove that Adv can also forge (∏i=1mH1(IDI,i)vi∗⋅g∑j=1nH2(IDI,j,τ∗)mj∗)xI.

That is,

(9) (σ^∗)xI=(∏i=1mH1(IDI,i)vi∗⋅g∑j=1nH2(IDI,j,τ∗)mj∗)xI=(∏i=1m(gδi⋅φ(h)ϵi)vi∗⋅g∑j=1nζj⋅mj∗)xI=g(∑i=1mδivi∗+∑j=1nζjmj∗)⋅(αI+s⋅βI)⋅φ(RI⋅ZβI)∑i=1mϵivi=RI(∑i=1mδivi∗+∑j=1nζjmj∗)⋅g(∑i=1mδivi∗+∑j=1nζjmj∗)⋅(s⋅βI)⋅φ(RI⋅ZβI)∑i=1mϵivi

Then, we let (∑i=1mδivi∗+∑j=1nζjmj∗) as μ, so,

(10) gs=((σ^∗)xIRIμ⋅φ(RI⋅ZβI)∑i=1mϵivi)(βIμ)−1

To solve gs, we must ensure that the following conditions are met: V∗ cannot be orthogonal to Zp∗.

ID∗=IDI.

From the above analysis, it can be seen that for chal to solve the co-CDH problem, the following conditions must be met simultaneously: the probability of Adv attack success cannot be negligible and gs can be solved. Here are the events where Adv wins the game: V∗ is not orthogonal to Zp∗, chal is not terminating the query as A, B, C, and the number of times Create Query, Ext Query, H1 Query, Q Query and Sign Query is Oc, Oe, OH1, Oq, and Os, respectively.

The success probability of chal is denoted as succchal:=Pr≥Pr(A∩B∩C)=Pr(C)⋅Pr(A∩B|C)=Pr(C)⋅Pr(A|C)⋅Pr(B|A∩C). Next, we first calculate Pr(A|C) and Pr(B|A∩C) in Part 1, and Pr(C) is calculated in Part 2.

Part 1. Under the condition that event C occurs, event A occurring means that Adv has won the game without chal’s awareness. Assuming that this probability cannot be negligible, this probability is denoted as succAdv:=Pr(A|C).

Claim. Event A and event C occur simultaneously, and event B occurs as independent events. Then, Pr(B|A∩C)=Pr(B)=1−1p.

Proof. Event B is a complementary event of an Event that V∗ cannot be orthogonal to Zp∗, and the probability of the Event that V∗ cannot be orthogonal to Zp∗ occurring is the probability of an Event that ∑i=1mδivi∗+∑j=1nζjmj∗=0 occurring; that is, Pr(B¯)=1p, then Pr(B|A∩C)=Pr(B)=1−Pr(B¯)=1−1p.

Part 2. The occurrence of event C can be divided into three steps: First, ensure that Ext Query does not terminate the query, then ensure that Q Query does not terminate the query, and finally, ensure that Sign Query does not terminate the query.

The probability of not terminating the query in Ext Query is (1−1Oc)Oq, the probability of not terminating the query in Sign Query is (1−1OcOH1)Os, and the probability of not terminating the query in Q Query is (1−1Oc)Oq. Then, Pr(C)=(1−1Oc)Oe⋅ (1−1OcOH1)Os⋅(1−1Oc)Oq=(1−1Oc)Oe+Oq⋅(1−1OcOH1)Os.

In sum, succchal≥(1−1Oc)Oe+Oq⋅(1−1OcOH1)Os⋅(1−1p)⋅succAdv>12λ means that chal can solve the co-CDH problem with a non-negligible probability through a PPT algorithm, which contradicts the co-CDH assumption. This proves that there is no such challenger, in turn proving that there is no such adversary. The signature scheme achieves (t,q,ϵ)-security in the EUF-CMA model.

Confidentiality proof

In this section, we will prove the confidentiality of the scheme under adaptive chosen message attacks in the random oracle model. We use the challenge-response mode for the proof. Let the challenger be denoted as C, the adversary as A, and the number of queries to the random oracle as q.

Initialization. The challenger C initializes the public parameters and master secret key as described in the initialization phase.

Random oracle queries. The adversary A adaptively queries the random oracle OH. The challenger responds to these queries as follows: For each query H(m), if m is already in the query table, return the stored value. Otherwise, randomly select a value from G1 and return it, storing the result in the query table.

For each query to the vector value, if the vector is already in the query table, return the stored value. Otherwise, randomly select a value and return it, storing the result in the query table.

For each query to the session key, if the session key is already in the query table, return the stored value. Otherwise, randomly select a value and return it, storing the result in the query table.

Signature queries. The adversary A can request signatures on chosen messages. The challenger responds by generating the signatures as described in the signature phase.

Output phase. The adversary A outputs a guess for the ID associated with the message. The challenger C checks if the guess is correct. If it is, the adversary wins the game.

The probability of the adversary A winning the game is analyzed as follows: Let A be the event that A wins the game.

Let B be the event that the A’s query is orthogonal to the challenge.

The probability of the adversary winning the game is given by:

(11) Pr(A)=Pr(A|B)⋅Pr(B)+Pr(A|B¯)⋅Pr(B¯)

Since Pr(B) is negligible, and Pr(A|B¯) is also negligible, we can conclude that Pr(A) is negligible. Thus, the scheme is confidential under adaptive chosen message attacks in the random oracle model.

Unlinkability proof

Game definition

1. Setup: The challenger generates params and msk, and registers two users U0 and U1 with private keys uskID0 and uskID1.

2. Challenge phase: The challenger randomly selects b∈{0,1}, uses uskIDb to generate a signature σ∗, and sends σ∗ to the adversary.

3. Adversary queries: The adversary can request signatures for other messages or users (excluding U0 and U1) and perform verifications.

4. Guess: The adversary outputs a guess b′. The scheme is unlinkable if |Pr[b′=b]−12|≤negl(λ).

Key observations

Private key randomness: Each user’s uskID=(x,R) includes a unique random r in R=hr. Since r is fresh per user, x=r+s⋅H0(ID,R) is statistically independent across users.

Signature randomization: Signatures σk=σ^kx⋅H3(vk,ID)BmodQ depend on both x (user-specific) and BmodQ (session-specific). The term H3(vk,ID)BmodQ introduces session randomness, preventing linkage across different signatures.

Session key obfuscation: The CRT-based distribution of bID=BmodqID ensures that partial knowledge of {bIDj} does not reveal B unless t combiners collude. This threshold mechanism hides user-specific contributions.

Formal reduction

Assume an adversary A can win the unlinkability game with non-negligible advantage ϵ. We construct a solver S for the CDH problem: 1. S embeds a CDH instance (g,h,ga,hb) into the public parameters and simulates user keys using a,b.

2. When A requests a signature, S programs the hash oracles to align with the CDH challenge.

3. If A successfully links signatures, S extracts e(g,h)ab from the bilinear pairing results, solving CDH.

4. By the CDH assumption, ϵ must be negligible, contradicting A’s advantage. Hence, unlinkability holds.

Critical analysis

Leakage prevention: No phase reveals s, B, or deterministic relationships between users’ operations. The use of fresh randomness ( r, B) in key generation and signing ensures unlinkability.

Verification anonymity: The verification equation e(ςk,h)=e(σ^k,R⋅mpkH0(ID,R)) depends only on public values ( R,mpk) and session-specific terms, avoiding user identity exposure.

Threshold security: The requirement of t combiners to recover B ensures that fewer colluders cannot compromise session anonymity.

The scheme achieves unlinkability through: Randomized private key generation and session-specific parameters.

Threshold-based session key distribution via CRT.

Cryptographic primitives (bilinear maps, collision-resistant hashes) that prevent leakage of user-specific information.

Dynamic binding of signatures to session-specific terms rather than user identities.

Under the challenge-response model, the adversary cannot distinguish signatures from different users beyond random guessing, proving the scheme’s unlinkability.

Traceability proof

Game definition

1. Setup: The challenger generates params, msk, and registers a set of users U. Each user Ui receives uskIDi=(xi,Ri).

2. Adversary queries: The adversary can: Request user private keys for any IDj∈U.

Request signatures on messages with specified IDj.

Corrupt combiners to obtain their session keys {bIDj}.

3. Challenge: The adversary outputs a forged signature σ∗ on a message m∗, claiming it cannot be traced to any registered user.

4. Tracing: The challenger uses the tracing algorithm to extract an identity ID∗ from σ∗. The scheme is traceable if Pr[ID∗∈U∧Verify(m∗,σ∗)=1]≥1−negl(λ).

Key mechanisms for traceability

Identity binding in private keys: Each user’s uskID=(x,R) is bound to ID via x=r+s⋅H0(ID,R). The term H0(ID,R) ensures that x uniquely encodes ID, and any valid signature must use a valid x linked to a registered identity.

Signature structure: Signatures σk=σ^kx⋅H3(vk,ID)Bmod Q explicitly include ID in H3. During verification, the challenger can check the consistency of ID with the public parameters and traced keys.

Session key recovery via CRT: The threshold-based recovery of B requires at least t honest combiners. If a forged signature uses an invalid B, the tracing algorithm can identify corrupt combiners by analyzing inconsistencies in B′.

Formal reduction

Assume an adversary A can forge an untraceable signature with non-negligible probability ϵ. We construct a solver S for the DLP in G1: 1. S simulates the scheme and embeds a DLP instance h=ga into the public parameters.

2. When A requests a signature for IDj, S programs H0(IDj,Rj) to align with the DLP challenge.

3. If A outputs a forged σ∗, S extracts x∗ from σ^k via Eq. (12), (12) e(σ^k,h)=e(gx∗,h)⇒x∗=loggσ^k

since x∗=r+s⋅H0(ID∗,R∗), S solves a from h=ga using the extracted x∗.

4. By the DLP assumption, ϵ must be negligible, contradicting A’s success. Hence, traceability holds.

Critical analysis

Non-frameability: Even if the adversary corrupts users, they cannot forge signatures for honest users because xi depends on s (unknown to the adversary).

Threshold security: The CRT-based recovery of B ensures that corrupting fewer than t combiners does not compromise B, preventing fake session keys from being accepted.

Public verifiability: The verification equation e(ςk,h)=e(σ^k,R⋅mpkH0(ID,R)) ensures that only valid ID-bound signatures pass verification.

The scheme achieves traceability through: Cryptographic binding of user identities to private keys via H0(ID,R).

Explicit inclusion of ID in signature components and hash functions.

Threshold mechanisms for session key recovery, limiting collusion impact.

Under the challenge-response model, any forged signature can be traced to a registered user with overwhelming probability, proving the scheme’s traceability.

Coalition-resistance proof

Phase 1: initialization and user registration

Challenger’s operations: - Generates public parameters params and master private key msk=s.

- For each user identity IDi requested by the adversary, generates private key uskIDi=(xi,Ri), where xi=ri+s⋅H0(IDi,Ri) and Ri=hri.

Collusion resistance analysis: - Even if the adversary obtains multiple users’ private keys {uskIDi}, they cannot recover s through linear combinations due to the independence of ri and the secrecy of s (relies on the hardness of the discrete logarithm problem).

Phase 2: combiner permission application

Challenger’s operations: - Generates session keys bID=B mod qID for legitimate combiners and distributes them via CRT.

Collusion resistance analysis: - If the adversary controls t′ combiners with {bIDj,qIDj}j=1t′ and t′<t (threshold), B cannot be recovered (CRT requires at least t pairwise coprime qID).

- If the adversary forges qID or bID, the verification in Phase 4 detects B′≠B and terminates the protocol.

Phase 3: signature generation and distribution

Challenger’s operations: - Generates signatures σk=σ^kx⋅H3(vk,ID)BmodQ for data.

Collusion resistance analysis: - To forge a signature, the adversary must compromise both x and B: ∗ x is protected by user private keys and bound to s.

∗ B is distributed via CRT; partial knowledge of bID is insufficient for recovery (relies on CRT security).

Phase 4: verification and combination

Challenger’s operations: - Verifies e(ςk,h)=e(σ^k,R⋅mpkH0(ID,R)) during validation.

Collusion resistance analysis: - If colluders tamper with σk or v~d, the bilinear pairing check fails (relies on collision resistance of hash functions and properties of bilinear maps).

Assume an adversary A can break collusion resistance with non-negligible probability ϵ. We construct an algorithm C′ to solve the CDH problem: 1. C′ simulates the scheme using A’s queries and embeds a CDH instance into public parameters.

2. When A forges a signature, C′ extracts the CDH solution from the bilinear map result.

3. By the CDH assumption, ϵ is negligible, leading to a contradiction. Hence, collusion resistance holds.

The scheme achieves collusion resistance through: Master key protection: Distributed generation and verification of s and B, preventing single-point leakage.

Session key distribution: Threshold mechanism based on CRT, requiring colluders to exceed the security threshold.

Cryptographic primitives: Bilinear map verification and collision-resistant hash functions ensure tamper detection.

Dynamic binding: Signatures are bound to file identifiers τ, preventing replay attacks.

Thus, under the challenge-response model, the scheme resists collusion attacks.

Performance analysis

The performance of the proposed scheme is analyzed from both theoretical, experimental simulation and perspectives. The detailed contents are presented in the subsequent sections.

Theoretical analysis

Given that the restricted group of combiners is the first construction proposed in this article and is not directly comparable to other schemes, we focus our analysis on comparing the signature and verification efficiencies of our scheme with those presented in Boneh et al. (2009), Lin et al. (2018), Chang et al. (2019), Lin, Xue & Huang (2021), and Li, Zhang & Liu (2022). In efficiency analysis, the specific parameters are made in Table 3.1 All computational costs are derived directly from the algorithms section of each article, by summing the operations specified in both the signature and verification expressions. The concrete efficiency analysis is shown in Table 4.

Table 3 Notations.

Notation	Meaning	
|G1|	Element size on group G1	
|G2|	Element size on group G2	
|Zp|	Element size on group Zp	
TM1	Multiplication operations on the group G1	
TM2	Multiplication operations on the group G2	
TM3	Multiplication operations on the group GT	
TE1	Exponential operations on the group G1	
TE2	Exponential operations on the group G2	
TE3	Exponential operations on the group GT	
TH1	Mapping to the hash operation on group G1	
TH2	Mapping to the hash operation on group G2	
TP	bilinear pairing operation	
m	Number of vectors	
n	Number of dimensions for each data vector	
c	Number of combined vectors	

Table 4 Efficiency analysis.

Scheme	Sig.comp	Ver.comp	AS	
Boneh et al. (2009)	(m+n+1)TE1+(m+n−1)TM1+mTH1	2cTP+(m+nc)TE1+(m+nc−1)TM1+mTH1	✓	
Lin et al. (2018)	(m+4)TE1+(m+1)TM1+(m+1)TH1	3cTP+(cm+cn+1)TE1+(cm+cn)TM1+(m+n+1)TH1+cTE3+cTM3	×	
Chang et al. (2019)	(m+n+3)TE1+(m+n+1)TM1+(m+n)TH1	3cTP+(cm+cn+1)TE1+(cm+cn)TM1+(m+n+1)TH1+cTE3+cTM3	×	
Lin, Xue & Huang (2021)	TP+(m+n+1)TE1+(m+n)TM1+(m+2)TH1	4cTP+(m+nc)TE1+(m+nc−1)TM1+(m+2c)TH1+cTE3+cTM3	✓	
Li, Zhang & Liu (2022)	(m+2)TE1+mTM1+mTH1	2cTP+(m+c)TE1+(m+c−1)TM1+mTH1+TE2+TM2	✓	
Our scheme	(m+3)TE1+(m+1)TM1+(m+1)TH1	2cTP+(m+2c)TE1+(m+2c−1)TM1+TE2+TM2	✓	
Note:

AS, using asymmetric bilinear pairing operations.

Table 4 shows that the signature overhead of all schemes, except for Lin et al. (2018), Li, Zhang & Liu (2022), and the proposed scheme, increases with the value of n. The proposed scheme reduces the number of TE1 exponential operations compared to the (Lin et al., 2018) scheme by extending its functionality. In terms of verification overhead, all schemes, except Li, Zhang & Liu (2022) and the proposed scheme, also experience an increase as n grows. Although the proposed scheme has c(TE1+TM1)−mTH1 operations more than the (Li, Zhang & Liu, 2022) scheme, the performance degradation is minimal. Overall, our analysis indicates that the proposed scheme maintains effective signature and verification efficiency while enhancing the original scheme’s functionality.

Experiment analysis

The computational cost of this scheme is calculated through computer simulation. Our simulation uses the Pycharm IDE, with the compiler version being Python 3.8. Our simulation runs on a Linux 5.10.0-8-generic system, using the 11th Gen Intel (R) Core (TM) i7-11800H @ 2.30-GHz processor. The parameters used in the simulation are all standard parameters of the Pypbc library (Duquesne et al., 2006). To design an 80-bit secure2 signature algorithm, the length of any element in the group G1 needs 160 bits (20 bytes), and the length of any element in the group G2 needs 320 bits (40 bytes). The length of each parameter during the experiment is shown in Table 5. In this experiment, we selected a 13-KB file to evaluate the efficiency of each scheme. Each file can be divided into m basic vectors, each basic vector can be divided into n dimensions, and the augmented vectors’ dimension is N=m+n. The basic vectors of each dimension are one element of Zp, with a length of 160 bits. For security reasons, the combiners number t∈[⌈m3⌉,m]. Therefore, 1608(m−1)n≤13Kt≤1608mn. For most of the current IP networks, the maximum transmission unit (MTU) payload is 1,460 bytes, so m+n≤73. Following comprehensive consideration, we can conclude that m+n∈[52,73]. Our scheme was compared with the aforementioned five schemes under the conditions of t=⌈m3⌉ and t=m, as detailed in Figs. 5 to 8.

Table 5 The length of elements.

Parameter	Length	
The element of G1	160 bits	
The element of G2	320 bits	
The element of GT	640 bits	
The element of H0	160 bits	
The element of H1	160 bits	
The element of H2	160 bits	
The element of H3	160 bits	

Figure 5 The time cost of signature generation for a file (t = ⌈m/3⌉).

Figure 6 The time cost of signature generation for a file(t = m).

Figure 7 The time cost of data packet detection(t = ⌈m/3⌉).

Figure 8 The time cost of data packet detection(t = m).

Figures 5 and 6 show our scheme compared to the other five schemes, as well as the signature computing cost of the same file. In all cases, the signature of the generating cost for our scheme and Li, Zhang & Liu (2022) are the smallest, and our proposed scheme also has the ability to limit the scope of combiners. When the basic vectors’ dimension rises, Lin et al. (2018), Li, Zhang & Liu (2022) and our scheme exhibit a gradual drop in the signing time drop, while the signing time of the other three schemes continuously rises. At the maximum transmission efficiency, when the transmission packet size reaches the maximum transmission unit (MTU) with parameters n=62, m=11, the signature cost of our scheme is 27 ms. This is nearly equivalent to the 24 ms reported in Li, Zhang & Liu (2022) and is significantly lower than the costs associated with the other four schemes.

In this simulation experiment, we assume the receiver receives c=2 packets. As illustrated in Figs. 7 and 8, when compared with the other five schemes, the time cost of our scheme and that of Li, Zhang & Liu (2022) remains minimal as the dimensionality of the basic vectors increases. When the transmitted data packets reach MTU, the verification cost of our scheme is 44 ms. This is nearly equivalent to the 39 ms reported in Li, Zhang & Liu (2022) and is significantly lower than the costs associated with the other four schemes.

Comprehensive analysis shows that when the file size reaches the MTU, our scheme has equivalent time overhead to Li, Zhang & Liu (2022). We provided a formal description of the definition and security model of IBLHS-RCG and proposed an efficient multiparty supervision scheme suitable for electronic financial transactions. Our scheme can be used to designate combinations to prevent malicious combinations. We demonstrated the security of our scheme. In addition, theoretical analysis and experimental results indicate that our scheme is more efficient and reasonable than comparable schemes.

Conclusion

In this article, we provide a formal definition and security model for the identity-based linear homomorphic signature for a restricted combiners’ group (IBLHS-RCG) and propose an efficient multiparty supervision scheme designed specifically for electronic financial transactions. Our scheme allows for the careful designation of combinations to reduce the risk of malicious collaborations. We establish the security of our approach, and both theoretical analysis and experimental results show that our scheme offers superior efficiency and practicality compared to existing methods.

Supplemental Information

Supplemental Information 1 Original data of Figure 5.

Supplemental Information 2 Original data of Figure 6.

Supplemental Information 3 Confidentiality Proof & Coalition-resistance Proof.

Supplemental Information 4 Unlinkability Proof & Traceability Proof.

Supplemental Information 5 Simulation code.

The computational cost of this scheme is calculated through computer simulation. Our simulation uses the Pycharm IDE, with the compiler version being Python 3.8. Our simulation runs on a Linux 5.10.0-8-generic system, using the 11th Gen Intel (R) Core (TM) i7-11800H @ 2.30-GHz processor. The parameters used in the simulation are all standard parameters of the Pypbc library.

Additional Information and Declarations

Competing Interests

The authors declare that they have no competing interests.

Author Contributions

Yuan Tian conceived and designed the experiments, performed the experiments, analyzed the data, performed the computation work, prepared figures and/or tables, authored or reviewed drafts of the article, and approved the final draft.

Weitao Song conceived and designed the experiments, prepared figures and/or tables, and approved the final draft.

Tanping Zhou conceived and designed the experiments, prepared figures and/or tables, authored or reviewed drafts of the article, and approved the final draft.

Bin Hu conceived and designed the experiments, prepared figures and/or tables, and approved the final draft.

Xuan Zhou conceived and designed the experiments, prepared figures and/or tables, authored or reviewed drafts of the article, and approved the final draft.

Yujie Ding conceived and designed the experiments, authored or reviewed drafts of the article, and approved the final draft.

Weidong Zhong conceived and designed the experiments, authored or reviewed drafts of the article, and approved the final draft.

Xiaoyuan Yang conceived and designed the experiments, authored or reviewed drafts of the article, and approved the final draft.

Data Availability

The following information was supplied regarding data availability:

The code and raw data are available in the Supplemental Files.

1 The method used to map the hash function onto a group (i.e., H1 and H3) first uses a hash function to calculate the hash value of the input into the algorithm as the elliptic curve’s abscissa x; this value was then brought into the corresponding elliptic curve of the group to calculate the corresponding ordinate y. If there is not such y, continue to hash the abscissa x to obtain a new hash value as the abscissa x′, and this process is repeated until the corresponding y is found. The hash functions (i.e., H0 and H2) mapped to a finite field only need to map the input to the numerical values on the field. So, H1 and H3 are much less efficient than H0 and H2 (Boneh, Lynn & Shacham, 2004). Therefore, the computational cost of H1 and H3 in tH operations is ignored.

2 Refer to Lynn (2007) for detailed specifications regarding the parameters utilized in this study.

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
