# Peer review of "Identity-based linear homomorphic signature for a restricted combiners’ group for e-commerce"

_PeerJ Computer Science, doi:10.7717/peerj-cs.3068_

## Round 0.1 · original submission · Major Revisions

The referral process is now complete. The referees and I feel that more work could be done before the paper is published. My decision is therefore to provisionally accept your paper subject to major revisions. This does not mean that your paper will be accepted after a revision.

The referee comments are given below and on the website. Please make sure that you have addressed all the comments.

Reviewer 1 has suggested that you cite specific references. You are welcome to add it/them if you believe they are relevant. However, you are not required to include these citations, and if you do not include them, this will not influence my decision.

**Language Note:** The review process has identified that the English language must be improved. PeerJ can provide language editing services - please contact us at [email protected] for pricing (be sure to provide your manuscript number and title). Alternatively, you should make your own arrangements to improve the language quality and provide details in your response letter. – PeerJ Staff

Reviewer 1 ·

Basic reporting

This paper proposes a novel homomorphic signature scheme where aggregation of individual signatures is restricted within an authentic group. The proposed technology uses the concept of identity based cryptography to avoid the problem of certificate management, and implements using bi-linear map on elliptic curves. The paper is clearly written and well structured. Number of references are adequate.

Experimental design

The authors have analytically computed the execution costs of their proposed algorithms and compared with two existing related schemes. The authors have also measured the execution times of each individual cryptographic operations on a specific hardware platform.

Validity of the findings

Random oracle model based formal verification done by the authors are sound and hence establishes the security properties.

Additional comments

Authors should clarify more clearly why they have compared with the scheme of Lin et al. (2021) only in the introduction section? A tabular comparison with the other existing schemes in wither the related work section or analysis section would be more convincing for the readers.

·

Basic reporting

a. Clear and unambiguous, professional English used throughout.
This search requires minor proofreading. The level of English writing is good.
b. Literature references, sufficient field background/context provided.
Literature references and background/context are presented appropriately, however, it requires addition more references.
c. Professional article structure, figures, tables. Raw data shared.
The structure of the paper is mostly straightforward. However, there are some questions that we added in our comments about the proposed scheme (in Figure 3) that require careful addressing. In addition to Figure 4, its results require clarification of the size of the keys used and the length of the hash, because this greatly affects the results.
d. Self-contained with relevant results to hypotheses.
We think it's appropriate.
e. Formal results should include clear definitions of all terms and theorems, and detailed proofs.
There is some uncertainty about the results of Figure 4 and even Figure 5. Authors should carefully explain all parameters used and their values.

Experimental design

a. Original primary research within Aims and Scope of the journal.
This research is within Aims and Scope of the journal.
b. Research question well defined, relevant & meaningful. It is stated how research fills an identified knowledge gap.
Research question is implicitly well defined. Knowledge gap is mostly defined and clear in this paper.
c. Rigorous investigation performed to a high technical & ethical standard.
This research did provide rigorous investigation performed for both high technical and ethical standard.
d. Methods described with sufficient detail & information to replicate.
In this research, methods are mostly clear and are well described. However, it requires some improvements (as mentioned in our comments).

Validity of the findings

a. Impact and novelty not assessed. Negative/inconclusive results accepted. Meaningful replication encouraged where rationale & benefit to literature is clearly stated.
We believe that the evaluation of the security analysis was good, but the performance evaluation requires improvement.
b. All underlying data have been provided; they are robust, statistically sound, & controlled.
This paper does not include underlying data.
c. Conclusions are well stated, linked to original research question & limited to supporting results.
Conclusions are not well stated and require improvements.
d. Speculation is welcome, but should be identified as such.
This paper does not contain speculation.

Additional comments

The author should carefully address all our concerns below:
- Why do some paragraphs in the introduction contain no references?
- The authors indicated that “to prevent unauthorized sub-servers from arbitrarily combining" line 71, Do these servers contain copies of user information for e-commerce applications?
- Why did the authors rely on Lin's scheme to share secrets instead of Shamir's scheme? Authors should specify clear reasons for the paper.
- In Figure 3, how will the combiner distinguish a specific signer? It is not clear.
- Also in Figure 3, when the ID is transferred explicitly from the receiver to all network entities including the singer, isn't this considered a data leak!!!
- In addition, when four hashes (H0, H1, H2 and H3) are used in each signature without specifying whether it is lightweight or standard, do these operations not increase the complexity of the scheme and decrease performance?
- It is not clear how signing keys between signers, combiners and receivers are confidentially updated.
- In Figure 4, how did the results of the proposed scheme appear without mentioning what are the lengths of the keys used and what is the length of the message digest of the hash used in the proposed scheme?
- Acronyms must define before using in the first appearance such as EUF, CMA, CDH, PKG etc.
- Some recent references deal with signatures and the homomorphic property on the one hand, and e-commerce applications and tracking data leaks on the other hand. They may be useful for references.
o https://www.mdpi.com/2076-3417/10/6/2007
o https://link.springer.com/chapter/10.1007/978-981-99-3588-8_5
o https://www.mdpi.com/1999-5903/15/8/262
o https://dl.acm.org/doi/abs/10.1145/3603781.3604219
- English writing: This paper needs minor proofreading to make it free of mistakes.
- References list: The references must follow PeerJ Computer Science style. A large number of references are outdated and require updating. All journal names should be in italics. Some references do not contain enough information. The authors should carefully and accurately check all references.

---

## Round 0.2 · Minor Revisions

The paper does not present a detailed comparison. The literature review is not complete. Recent and important references are missing. There are also significant problems with the organization. Therefore, the decision is subject to minor revision.

·

Basic reporting

a. Clear and unambiguous, professional English used throughout.
The level of English writing is good.
b. Literature references, sufficient field background/context provided.
Literature references and background/context are presented appropriately.
c. Professional article structure, figures, tables. Raw data shared.
The structure of the paper is mostly straightforward.
d. Self-contained with relevant results to hypotheses.
We think it's appropriate.
e. Formal results should include clear definitions of all terms and theorems, and detailed proofs.
We think it's appropriate.

Experimental design

a. Original primary research within Aims and Scope of the journal.
This research is within Aims and Scope of the journal.
b. Research question well defined, relevant & meaningful. It is stated how research fills an identified knowledge gap.
Research question is implicitly well defined. Knowledge gap is mostly defined and clear in this paper.
c. Rigorous investigation performed to a high technical & ethical standard.
This research did provide rigorous investigation performed for both high technical and ethical standard.
d. Methods described with sufficient detail & information to replicate.
In this research, methods are mostly clear and are well described.

Validity of the findings

a. Impact and novelty not assessed. Negative/inconclusive results accepted. Meaningful replication encouraged where rationale & benefit to literature is clearly stated.
We believe that the evaluation of the security analysis was good.
All underlying data have been provided; they are robust, statistically sound, & controlled.
This paper does not include underlying data.
b. Conclusions are well stated, linked to original research question & limited to supporting results.
We think it's appropriate.
c. Speculation is welcome, but should be identified as such.
This paper does not contain speculation.

Additional comments

The authors responded to all our comments carefully.
Thanks

Reviewer 3 ·

Basic reporting

In this paper, the authors proposed a new homomorphic signature scheme to achieve hierarchical openness in electronic transactions. By using a CRT-based multiparty composable signature algorithm, the constructed idea is designed for IBLHS-RCG.
• The abstract is not well written. The authors should rewrite it by giving basic details about the method and novelty of the paper.
• The introduction part needs to be rewritten. It should explain the importance of the homomorphic signature requirement of e-commerce.
• The author provides information on financial transactions in the first part of the manuscript. However, this does not give the current state of the art.
• The organization of the introduction section is not well defined. The authors explained some concepts in that part and gave some literature. They should build a connection between the ideas. By making the connection, they should feature the importance of the proposed idea.
• The contributions section is not well-defined. What is the main contribution of this paper to literature? By proposing this scheme, what open problem is solved? Clarify them.
• The authors should also add some parts to highlight the main motivation of the paper.
• The language is terrible. It needs proofreading.
• The related work section does not cover the current state of the art. By systematically analyzing the existing literature, the authors should update the related work section.
• In the preliminaries, a notation table should be added, and security-related definitions should be given in another subsection.

Experimental design

• The formal definition of IBLHS-RCG should be given with the protocol flow structure to make understanding more transparent. In other words, the proposed method should be explained by providing an algorithm, pseudocode, or something like that.
• The entire workflow of the scheme, given in Figure 3, should be detailed by adding a step-by-step explanation.
• In the experiment analysis section, the source codes should be given, and the time cost examinations should be delivered in a table by providing a comparison between literature solutions.
• They should provide detailed evidence that the proposed scheme is suitable for electronic financial transactions by giving mathematical proof and experimental analysis.

Validity of the findings

• The correctness of the proposed scheme is unclear. Firstly, the authors should define how a homomorphic signature can be verified. Then, by looking at the proposed idea, they should rewrite the correctness analysis.
• The authors should add a new section to discuss the literature. Then, they also need to explain the main differences of the proposed idea from the literature.
• The proposed homomorphic signature idea is not well defined.
• Based on the proposed idea, the authors should give more detail about security analysis. Firstly, they should define the following security model. After that, they have to analyze the security of the proposed signature by measuring the advantage of the adversary according to the random oracle model.

Additional comments

• The abstract and conclusion do not describe the content/scope of the paper. The methods expressed in the paper should be summarized comprehensively. The advantages and shortcomings should be briefly summarized.
• Figures 4-5 make no sense for practical use cases.
• In my opinion, the most critical shortcoming of the study is the lack of meaningful comparison results, detailed explanation of the proposed idea, and limited contribution.
• This paper is not ready for publication. Much more effort is needed.

Annotated reviews are not available for download in order to protect the identity of reviewers who chose to remain anonymous.

---

## Round 0.3 · Major Revisions

The referral process is now complete. While finding your paper interesting and worthy of publication, the referees and I feel that more work could be done before the paper is published. My decision is therefore to provisionally accept your paper subject to major revisions.

·

Basic reporting

a. Clear and unambiguous, professional English used throughout.
The level of English writing is good.
b. Literature references, sufficient field background/context provided.
Literature references and background/context are presented appropriately.
c. Professional article structure, figures, tables. Raw data shared.
The structure of the paper is mostly straightforward.
d. Self-contained with relevant results to hypotheses.
We think it's appropriate.
e. Formal results should include clear definitions of all terms and theorems, and detailed proofs.
We think it's appropriate.

Experimental design

a. Original primary research within Aims and Scope of the journal.
This research is within Aims and Scope of the journal.
b. Research question well defined, relevant & meaningful. It is stated how research fills an identified knowledge gap.
Research question is implicitly well defined. Knowledge gap is mostly defined and clear in this paper.
c. Rigorous investigation performed to a high technical & ethical standard.
This research did provide rigorous investigation performed for both high technical and ethical standard.
d. Methods described with sufficient detail & information to replicate.
In this research, methods are mostly clear and are well described.

Validity of the findings

a. Impact and novelty not assessed. Negative/inconclusive results accepted. Meaningful replication encouraged where rationale & benefit to literature is clearly stated.
We believe that the evaluation of the security analysis was good.
All underlying data have been provided; they are robust, statistically sound, & controlled.
This paper does not include underlying data.
b. Conclusions are well stated, linked to original research question & limited to supporting results.
We think it's appropriate.
c. Speculation is welcome, but should be identified as such.
This paper does not contain speculation.

Additional comments

The paper in its current form is fit for publication, and the authors responded to all of our comments.

Reviewer 3 ·

Basic reporting

In this paper, the authors proposed a new homomorphic signature scheme to achieve hierarchical openness in electronic transactions. By using a CRT-based multiparty composable signature algorithm, the constructed idea is designed for IBLHS-RCG.

• The language is terrible. It needs proofreading.
• In my opinion, the authors tried to paraphrase some definitions or primitives by looking at some literature papers. However, they are not successful, and the structure of the manuscript is incomprehensible.
• Lines 275-280 are not acceptable without any concrete proof.

• The following articles can be useful to rebuild the proposed idea:
o https://www.techscience.com/CMES/v136n2/51581
o https://ieeexplore.ieee.org/document/9268072
o https://ieeexplore.ieee.org/document/8302552
o https://ieeexplore.ieee.org/abstract/document/8676234/

Experimental design

• The authors should add a new section to discuss the literature. Then, they also need to explain the main differences between the proposed idea and the literature by comparing it with previous studies.

• The concrete efficiency analysis is provided in Table 2. However, it does not give a meaningful comparison. How can we ensure that the proposed idea is better than the others? A concrete analysis needs to be constructed based on the length of the elements. Is there any other scheme between 2009 and 2021 to extend the comparison? I think Lin et al (2018) should also added.
• According to the authors, the proposed scheme is an identity-based signature. What kind of advantages can be obtained thanks to this property? Why should we choose the proposed scheme instead of Lin et al. (2021) ?

Validity of the findings

• The security analysis is missing. The unforgeability of the proposed scheme against the Adaptive Chosen Message Attack is not well-defined. Please check “Goldwasser, S., & Bellare, M. (1996). Lecture notes on cryptography. Summer course “Cryptography and computer security” at MIT, 1999, 1999.” and reanalyze the security of the proposed scheme by considering the adversary’s ability to produce the new signature. The authors need to define a security model for the restricted combiners group. Then, they have to look deeply into security.
• The authors also check the security-related features such as confidentiality, unlikability, traceability, coalition-resistance, etc.
• The provided definitions, definitions 1 and 2, are not formal. They have to define how the proposed scheme presents Unforgeability against Adaptive Chosen Message Attack.
• Correctness analysis needs to be rewritten by showing Verify (v˜d,S˜d)=1.
• As a security model, the authors can follow the model which was defined by “Lin, Q., Yan, H., Huang, Z., Chen, W., Shen, J., & Tang, Y. (2018). An ID-based linearly homomorphic signature scheme and its application in blockchain. IEEE Access, 6, 20632-20640.” They have to evaluate the security of the proposed scheme by considering definition 4, which was given by Lin et. al. (2018).

Additional comments

• What is the security level used in comparing schemes? A proposed parameter set should be added to the revised manuscript.
• In my opinion, the authors tried to paraphrase some definitions or primitives by looking at some literature papers. However, they are not successful, and the structure of the manuscript is incomprehensible.
• Lines 275-280 are not acceptable without any concrete proof.
• Without changing the security analysis approach, giving detailed evidence to ensure the correctness, and analyzing the difference from existing literature, in my opinion, the paper is not ready for publication.

Annotated reviews are not available for download in order to protect the identity of reviewers who chose to remain anonymous.

---

## Round 0.4 · Major Revisions

The previous comments of R3 have not been addressed in detail. The contribution is not clear. So, my decision is that you still need to perform major revisions.

·

Basic reporting

a. Clear and unambiguous, professional English used throughout.
The level of English writing is good.
b. Literature references, sufficient field background/context provided.
Literature references and background/context are presented appropriately.
c. Professional article structure, figures, tables. Raw data shared.
The structure of the paper is mostly straightforward.
d. Self-contained with relevant results to hypotheses.
We think it's appropriate.
e. Formal results should include clear definitions of all terms and theorems, and detailed proofs.
We think it's appropriate.

Experimental design

a. Original primary research within Aims and Scope of the journal.
This research is within Aims and Scope of the journal.
b. Research question well defined, relevant & meaningful. It is stated how research fills an identified knowledge gap.
Research question is implicitly well defined. Knowledge gap is mostly defined and clear in this paper.
c. Rigorous investigation performed to a high technical & ethical standard.
This research did provide rigorous investigation performed for both high technical and ethical standard.
d. Methods described with sufficient detail & information to replicate.
In this research, methods are clear and are well described.

Validity of the findings

a. Impact and novelty not assessed. Negative/inconclusive results accepted. Meaningful replication encouraged where rationale & benefit to literature is clearly stated.
We believe that the evaluation of the security analysis was good.
All underlying data have been provided; they are robust, statistically sound, & controlled.
This paper does not include underlying data.
b. Conclusions are well stated, linked to original research question & limited to supporting results.
We think it's appropriate.
c. Speculation is welcome, but should be identified as such.
This paper does not contain speculation.

Additional comments

no comment

Reviewer 3 ·

Basic reporting

I think, my previous comments are not well-considered. There is also some other problems:

The understandability of Figure 3 is not acceptable. I can not trace the protocol to understand what is doing. They have to change their presenting style of the proposed scheme. The provided details in lines 267-344, should be summarized in a Figure that defines the operations, exchange parameters, etc.

The definition of CRT should be added, and it should be explained how this method can contribute to the proposed scheme.

All abbreviations should be given with their explanation in the first used place.

Electronic financial transaction usage should be detailed by giving how this scheme can be used in that area.

The contributions of the paper are inadequate.

Definitions 4, 5, 6, and 7 should be written in a theoretical way. Adding the definitions will not cover the main idea of the proofs of the scheme.

Experimental design

According to the authors, the proposed scheme is an identity-based signature. The answers to these questions, "What kind of advantages can be obtained thanks to this property? Why should we choose the proposed scheme instead of Lin et al. (2021) ?" should be given in the experimental analysis section.

The authors claim that "In addition, theoretical analysis and experimental results indicate that our scheme is more efficient and reasonable in the real world than comparable schemes." I am not sure how it could be possible. They have to prove their claim by providing a detailed analysis.

How will the scheme be more compatible with the IP network environment? Combining an identity-based signature makes it possible.

Validity of the findings

Providing Table 4 can be acceptable. However, there is no meaningful comparison in terms of signature and verification efficiency analysis. How we can ensure that the proposed scheme is much more efficient than others? By adding concrete parameters for m, n, c, etc., an approximate efficiency analysis should be provided. In addition, Table 4 should be reviewed to show how the proposed idea comes into for if we compare the others. The authors just put some equations and there is no examination on that analysis.

In the security analysis, Confidentiality and Coalition-resistance needs to be added.


The post-quantum resistance of the proposed scheme should be reviewed.

Unlinkability and Traceability proofs should also be detailed by analysing used parameters or computations.

The real-world examinations should be detailed. In the paper, ı couldn't find a real-world case scenario to say that the proposed scheme is much better than the others. In my opinion, simulation results do not cover all these issues.

The proofs are added incomprehensibly. I cannot follow them to ensure their correctness and security. They have to be written.

Additional comments

The conclusion and abstract were not well-written. By reading that part, I can not understand the proposed scheme's ideas, methods, or analysis results.

There are many punctuation mistakes in lines 61, 67, 402, 404, etc.

---

## Round 0.5 · Minor Revisions

The review process is now complete. While finding your paper interesting and worthy of publication, the referees and I feel that more work could be done before the paper is published. My decision is therefore to provisionally accept your paper subject to minor revisions.

Reviewer 4 ·

Basic reporting

The manuscript presents an identity-based homomorphic signature scheme and demonstrates technical soundness. However, certain aspects require attention:

- Punctuation and spacing issues are prevalent throughout the manuscript, affecting readability. In addition, in line 229, the manuscript states, "In Section 2, we demonstrate the correctness of our scheme." Since there is no numbering in the sections, please give references with the section names.

- Introduction section should better highlight the paper's contribution by clearly stating the advantages of homomorphic signatures.

Experimental design

- Given that the proposed scheme is identity-based, it would be beneficial for the authors to provide a concise overview of identity-based cryptographic systems in the introduction. This would help readers unfamiliar with the concept understand the foundational principles and the context of the proposed scheme.

- The manuscript introduces a restricted combiner's group, distinguishing it from the single combiner approach presented by Lin et al. (2021). It is recommended that the authors clearly define this structure and elaborate on its necessity and advantages. Such clarification would underscore the novelty and contribution of their work.

- While the proposed scheme is more costly in terms of performance and efficiency, it is crucial to highlight the specific differences and improvements over Lin et al.'s scheme.

Validity of the findings

- In the technical analysis section, the manuscript presents a comparison table detailing the signature and verification efficiency of the proposed scheme. While the notations are provided, it would enhance clarity if the authors included a brief explanation—particularly for the proposed scheme—on how these computational costs are derived.

Reviewer 5 ·

Basic reporting

The authors introduce a formal definition of a novel cryptographic primitive Identity-Based Linear Homomorphic Signature for a Restricted Combiners’ Group (IBLHS-RCG) which does not exist in prior literature. This indicates original theoretical work. They creatively apply the CRTs in combination with homomorphic signatures to restrict malicious combinations, which is a new approach to tackling misuse in multiparty settings.
I think the authors' proposal is acceptable, but many sides the work need improvement.

Experimental design

The scheme tailored to electronic financial transactions combines cryptographic rigor with real-world applicability, but more explanation needed to strengthen the author's contributions.

Validity of the findings

-All figure sizes must be respected with each others. For example Figure 1 symbols are larger than other symbols.
-Fig3. is not readable in print
- Do not use italic in anywhere in the paper. e.g. fig 4
-Fig 5, fig6 labels are not readable at all! These results must be clearly explained to strengthen the author's contributions.

Additional comments

section numbers missing. Authors make sure they meet with PeerJ writing regulations and references. A lot of texts are not readable. Authors must respect this.

---

## Round 0.6 · accepted · Accept

We are happy to inform you that your manuscript has been accepted for publication since the comments have been addressed.

Reviewer 5 ·

Basic reporting

The authors introduces a secure and efficient Identity-Based Linear Homomorphic Signature scheme with restricted combiners (IBLHS-RCG), using the Chinese Remainder Theorem to prevent malicious data combinations in electronic financial transactions, and demonstrates low computational cost through formal analysis and simulations.

The research is exicting for the research community, the authors improved the work.

Experimental design

The authors improved the results, they used simulations with Pycharm. I think this is acceptable.

Validity of the findings

The authors showed low computational cost through formal analysis and simulations.

Additional comments

Some bibliograpghy is outdated. The authors must use recent and respected libraries.